# Extracting Sign Language Articulation from Videos with MediaPipe

**Carl Börstell**

Dept. of Linguistic, Literary and Aesthetic Studies (LLE)

University of Bergen (UiB)

`carl.borstell@uib.no`

## Abstract

This paper concerns evaluating methods for extracting phonological information of Swedish Sign Language signs from video data with MediaPipe's pose estimation. The methods involve estimating i) the articulation phase, ii) hand dominance (left vs. right), iii) the number of hands articulating (one- vs. two-handed signs) and iv) the sign's place of articulation. The results show that MediaPipe's tracking of the hands' location and movement in videos can be used to estimate the articulation phase of signs. Whereas the inclusion of transport movements improves the accuracy for the estimation of hand dominance and number of hands, removing transport movements is crucial for estimating a sign's place of articulation.

## 1 Introduction

Sign languages – or, *signed languages* – are languages produced with gestures articulated in space and perceived visually or tactilely. Over 200 sign languages have been documented around the globe (Hammarström et al., 2022) but they are minoritized and under-researched. One challenge for quantitative research on sign languages is that they generally lack a conventionalized representation in a machine-readable form, such as phonetic transcription or orthography (see e.g., Miller, 2006; Frishberg et al., 2012; Crasborn, 2015). Following technological advances in computer vision, methods have emerged that allow a degree of form-based analysis of body movements, such as gesturing and signing, through human body pose estimation tracking of either real-time or pre-recorded video data (Pouw et al., 2020). Whereas most body pose tracking utilized in sign/gesture research used to involve either wearable devices (e.g., motion capture sensors) (Puupponen et al., 2015) or 3D cameras (e.g., Kinect) (Namboodiripad et al., 2016; Trujillo et al., 2019), thus requiring designated hardware, there are now pre-trained models that do human body pose estimation either real-time through a regular video camera or on pre-recorded video data, providing a cost-efficient alternative that has proven to be reliable in estimating human gesturing (Pouw et al., 2020). A popular tool for such analysis is OpenPose (Cao et al., 2017), which has been successfully applied in research on both sign language and gesture (Östling et al., 2018; Börstell and Lepic, 2020; Ripperda et al., 2020; Fragkiadakis et al., 2020; Fragkiadakis and van der Putten, 2021; Fragkiadakis, 2022). A tool that has become available more recently is Google's MediaPipe (Lugaresi et al., 2019), which similarly performs human body pose estimation of video data and outputs coordinates of landmarks (joints and anchor points such as eyes, nose and eyebrows).

### 1.1 Sign Language and Computer Vision

Previous research using OpenPose has shown that it can be used to pre-process and analyze gesture and sign language video data in terms of assessing movement (estimating articulation, holds and movement patterns) (Börstell and Lepic, 2020; Ripperda et al., 2020; Fragkiadakis et al., 2020; Fragkiadakis, 2022; Fragkiadakis and van der Putten, 2021), hand dominance (which hand is articulating more) and the number of hands involved in signing (one- vs. two-handed signs) (Östling et al., 2018; Börstell and Lepic, 2020), the place of articulation (the hands' position relative to the body) (Östling et al., 2018; Börstell and Lepic, 2020; Fragkiadakis, 2022) and even non-manual features (Kimmelman et al., 2020; Saenz, 2022). These are all basic properties of describing the form of signs and establishing the phonological structure of a sign language (Brentari, 2019). Defining the

start and end points of the sign articulation, excluding transport movements to and from the place of articulation, is crucial to delimit the articulation phase of a sign (Jantunen, 2015). Signs can be described as either one- or two-handed, generally evenly distributed in any sign language lexicon (Börstell et al., 2016), and two-handed signs can be further divided into *unbalanced* signs with a single active articulator (the dominant hand articulating on/by the non-dominant hand) vs. *balanced* signs, for which both hands articulate simultaneously (van der Hulst, 1996; Sandler, 2006; Crasborn, 2011). While hand dominance is generally associated with individual handedness (whether the signer is left- or right-handed), it is crucial to know which hand is dominant in one-handed and unbalance two-handed signs to establish the place of articulation, which in itself can be meaningful through iconic mappings, e.g., the head being associated with concepts relating to cognition (Börstell and Östling, 2017; Östling et al., 2018; Börstell and Lepic, 2020). The number of hands in signs has also been found to be iconically linked to plurality, such that two-handed signs are more likely to denote plural concepts (Lepic et al., 2016; Börstell et al., 2016; Östling et al., 2018).

## 1.2 Aims

In this paper, I evaluate methods of analyzing videos from the Swedish Sign Language online dictionary (Svenskt teckenspråkslexikon, 2023) with MediaPipe. The methods aim at extracting basic information about the articulation and sign form, which can aid quantitative research on sign languages relating to phonology and form–meaning mappings. Specifically, the aim is to evaluate methods for estimating the **articulation phase** of signs (§3.1), which can inform further analyses of sign form, and classifying signs as either left- or right-handed as **hand dominance** (§3.2) and one- or two-handed in terms of **number of hands** articulating (§3.3). Based on the hand dominance estimation and segmentation of the articulation phase, the sign's main **place of articulation** (§3.4) is estimated relative to the body.

## 2 Methodology

### 2.1 Retrieving and Processing Sign Videos

Using data from the Swedish Sign Language online dictionary (Svenskt teckenspråkslexikon, 2023) containing information about the hand dom-

inance, number of hands and sign location for the over 20,000 signs in the database, a subset of 1,292 non-compound signs was sampled to represent a diverse set of signers in the videos (including left- and right-handed signers) and different places of articulation. Non-compounds were selected to limit each sign to a single main place of articulation and avoid combination of multiple, phonologically different elements (cf. Lepic, 2015). The sampled signs were downloaded with the `signglossR` package (Börstell, 2022) and then analyzed with the Python (3.10.5) implementation of MediaPipe (`mediapipe` 0.8.10.1), together with OpenCV (`opencv-python` 4.6.0.66) and NumPy (`numpy` 1.23.1) (Harris et al., 2020). Each video is analyzed frame by frame using the pose model estimating major landmarks on the body, represented visually in Figure 2 using the one-handed sign TAXI. The sampled sign videos vary between 35 and 312 frames in total (mean = 83, *SD* = 33) – some videos are recorded in 50 frames per second (fps), others at 25 fps. Of the 1,292 sampled videos, 43 (3.3%) show left-handed signers, the rest right-handed signers, and 567 (43.9%) involve a one-handed sign (1h), whereas 725 (56.2%) are two-handed, of which 338 are unbalanced (2h unbalanced) and 387 are balanced (2h balanced). The distribution of places of articulation is shown in Table 1.[1]

| Location | *n* | *%* |
|---|---|---|
| head | 469 | 36.3% |
| torso | 184 | 14.2% |
| hand/arm | 397 | 30.7% |
| neutral | 155 | 12.0% |
| low | 87 | 6.7% |

Table 1: Places of articulation in sample.

### 2.2 Normalizing MediaPipe Outputs

A total of 107,955 frames from 1,292 videos were analyzed with MediaPipe. The output was further processed using R (4.2.2) and the packages `tidyverse` (Wickham et al., 2019), `pracma` (Borchers, 2022), `scales` (Wickham and Seidel, 2022), `slider` and (Vaughan, 2021), and graphics were created with packages `ggbeeswarm` (Clarke and Sherrill-Mix, 2017), `ggchicklet` (Rudis, 2022), `ggforce` (Ped-

---

[1]Locations are more fine-grained in the dictionary database, but are lumped into five major categories here.

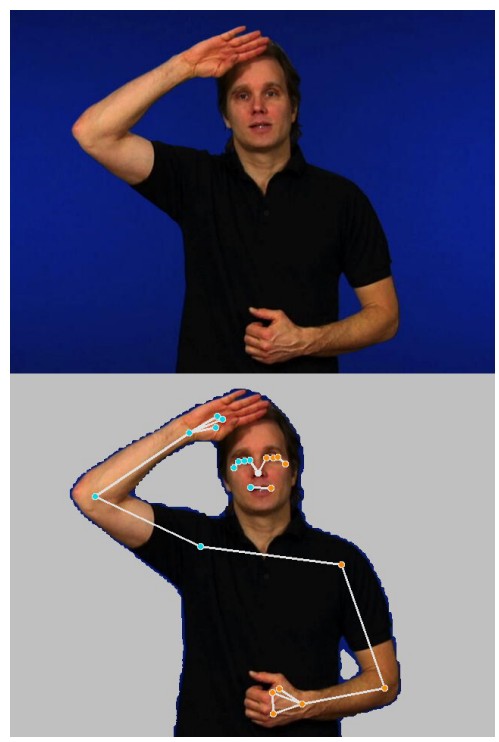

Figure 1: The sign TAXI (Svenskt tecken-språkslexikon, 2023, 1) (top) with the MediaPipe pose estimation visual output (bottom).

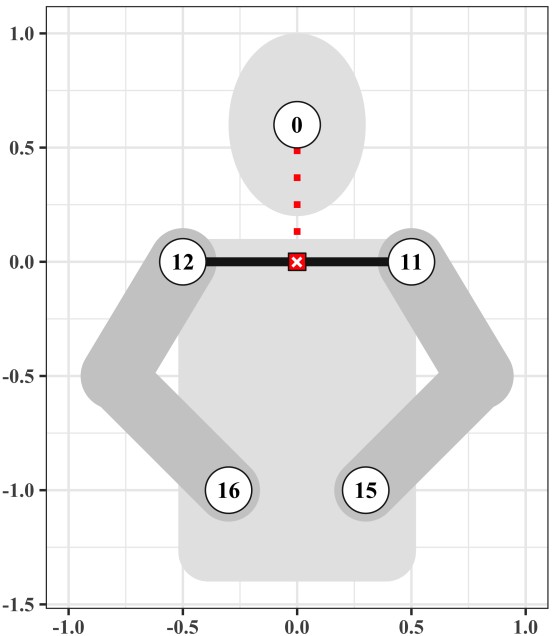

Figure 2: Relevant MediaPipe landmarks numbered, the normalized size based on the mean distance between shoulder landmarks scaled to 1, and origo set to the halfway point ("X" mark).

ersen, 2021), `ggrepel` (Slowikowski, 2022), `xtable` (Dahl et al., 2019).[2]

Only five out of the 33 landmarks of the pose estimation model were included in the further analysis, yielding a total of 539,775 datapoints, each representing a landmark estimation in a single frame. The five landmarks selected are shown in Figure 2: 0 represents the nose, 11 and 12 the left and right shoulders, and 15 and 16 the left and right wrists. The coordinate outputs from MediaPipe are scaled to 0 to 1 for both $x$ and $y$. Based on the methods of Östling et al. (2018) and Fragki-adakis and van der Putten (2021), coordinates are normalized based on the mean distance between the shoulders within a sign and adjusted to an origo set at the halfway point between the mean position of the two shoulders – the red square with a white "X" in Figure 2. The coordinates were rescaled such that the distance between the shoulders equals to 1 to normalize across signers of different size, and the distance between landmark 0 and origo equals .6, to approximate the proportions of the human body.

---

[2]The full data set and code can be found at: https://osf.io/x3pvq/.

## 2.3 Estimating Articulation

For each sign, the articulation phase was estimated based on the movement of the two hands (or, rather, wrists) represented by landmarks 15 and 16. For each hand, the Euclidean distance traveled between each frame transition was calculated and summed into a total distance traveled. The distance traveled was smoothed into a rolling average of ±2 frames. The smoothed distance traveled data was analyzed for peaks using the `pracma::findpeaks()` function, set to look for two peaks at least 8 frames apart. These peaks represent the highest points of articulation speed, assumed to occur to and from the articulation phase – i.e., transport movements. Then, the sequence between the two peaks identified was analyzed in isolation with the same function, but with inverted values to detect valleys – assumed to represent sign holds as onset/offset in syllables (Brentari, 2019) – and set to up to 6 peaks with at least 5 frames apart. The first (inverted) peak was defined as the start frame of the articulation phase, and the last (inverted) peak was defined as the end frame. If no (inverted) peaks were identified, the

start and/or end frames were defined as the first and last original (positive) peaks, respectively. If there were less than 10 frames between the start frame and the end frame, the end frame was extended to 10 frames after the start frame. Figure 3 illustrates the original signal of the total distance traveled by the hands in the sign TAXI in grey, the smoothed signal in black, with the identified peaks as vertical, black lines, and the inverted smoothed signal between peaks as a dashed, red line, with the inverted peaks identified as vertical, red lines.

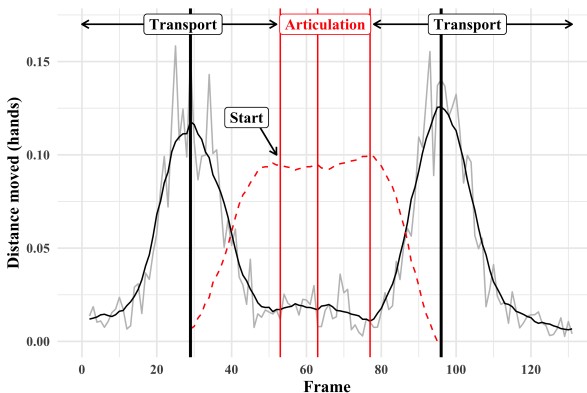

Figure 3: Distance traveled by the hands as a raw (grey) and smoothed (black) signal in the sign TAXI. Black lines show peaks in movement. The dashed, red curve is the inverted signal between peaks with lines representing peaks identified. First inverted peak is estimated start frame.

## 2.4   Estimating Hands

In order to estimate hand movements and locations reliably, it is important to establish which of the two hands is articulating in a sign, particularly for one-handed signs and unbalanced two-handed signs, for which the articulation is not symmetrical across the two hands. The estimation used here is simply comparing the distance traveled between the two hands: if the distance traveled by the right hand is equal to or greater than that of the left hand, the right hand is estimated to be the dominant hand, otherwise the left hand is estimated. This estimation is performed twice for each sign video: first with the distance traveled across all frames of the video (full method), then with the distance traveled within the estimated articulation phase only (short method).

Estimating the number of hands used in a sign is somewhat more complicated, as the relative difference in movement across the two hands can vary

a lot, especially when a non-articulating hand can still be moving because of general body motion or readjustments (changing rest position, grooming/scratching, etc.). Östling et al. (2018) used a factor of 3 as the cut-off point between one- and two-handed signs when analyzing sign language data with OpenPose: if one hand traveled over three times the distance of the other hand, the sign was estimated to be one-handed. However, one difference between the study by Östling et al. (2018) and this one is that they calculated an extrapolated position of the hands extended from the estimated wrist position, which could lead to differences in the distance traveled. In this paper, I evaluate the accuracy of different relative factors in the distance traveled by the two hands, ranging from 1 (equal distance) to 4 (four times the distance of the other hand). This estimation is also performed twice for each sign video: first with the distance traveled across all frames of the video (full method), then with the distance traveled within the estimated articulation phase only (short method).

The estimation of place of articulation is heavily dependent on an accurate classification of hand dominance, at least for one-handed signs. In this paper, the estimation of place of articulation is made on the basis of the location of the estimated dominant hand. Since several of the locations (see Table 1) are potentially overlapping and may display internal differences – e.g., signs articulated around the head may be high or low and right or left relative to the head – the main aim here is to estimate sign height, that is the location on the $y$ axis relative to origo. This estimation of place of articulation is done three times for each sign video: first using the mean coordinates of the estimated dominant hand across all frames of the video (full method), secondly, using the mean coordinates of the estimated dominant hand within the estimated articulation phase only (short method), and lastly using the coordinates of the estimated dominant hand of the estimated start frame only (start method).

## 3   Results

### 3.1   Articulation Phase

Using the peak estimation method on the distance traveled of the two hands, two main peaks were identified in all 1,292 sign videos. These peaks define the segment of the sign video that is fur-

ther analyzed for inverted peaks representing sign holds, when the hands are mostly stationary. For 47 (3.6%) out of 1,292 signs, no inverted peaks could be identified, in which case the original peaks were used as a proxy, and for 639 (49.5%) signs only a single inverted peak was found, in which case this is defined as the start frame. For 294 (22.8%) signs, the distance between start and end frames was less than 10 frames, resulting in the end frame being extended to 10 frames after the start frame. For the purpose of estimating place of articulation, the most important estimation is the initial hold phase at the beginning of the articulation phase, and with the current method of estimating this phase, 96.4% of the signs analyzed had an identified inverted peak between the transport movement peaks. Figure 4 illustrates the total distance moved by the hands across all sign videos, with vertical lines showing the mean relative locations of peaks and inverted peaks.

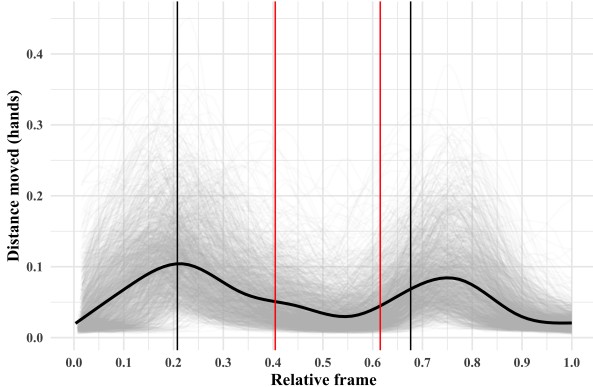

Figure 4: Distance traveled by the hands as a raw (grey) and smoothed (black) signal across all signs. Black lines show mean relative position of peaks in movement. The red lines show mean relative position of (inverted) peaks identified.

The accuracy of this method cannot be evaluated on its own without a manual annotation of each individual sign video's observed start and end points of the articulation phase. However, the method can be evaluated indirectly in the following sections, in terms of how useful the segmentation is for accurately estimating other form features of the signs, and the method will thus be discussed in more depth later.

## 3.2 Hand Dominance

The estimation of hand dominance was based on a simple comparison of the distance traveled by the left and right hands: if the distance traveled by the right hand is greater or equal to that of the left hand, the right hand was estimated to be the dominant hand – defaulting to the right hand for equal distances is motivated by the general right-handedness bias. The relative distance comparison was made across all frames (full method) and the frames within the estimated articulation phase only (short method).

Table 2 and Figure 5 show the accuracy of the two methods in classifying left- and right-dominant sign videos based on the actual handedness of the signers in the lexical database. The results show that the full method performs better than the short method, but both methods have a similar precision on left- and right-dominant signs.

| Method | Hand | Precision | Recall | $F_1$ |
|--------|------|-----------|--------|-------|
| Full | left | 0.81 | 0.88 | 0.85 |
| Full | right | 0.81 | 0.81 | 0.81 |
| Short | left | 0.72 | 0.72 | 0.72 |
| Short | right | 0.72 | 0.72 | 0.72 |

Table 2: Precision, recall and $F_1$ of hand dominance estimation with full and short methods.

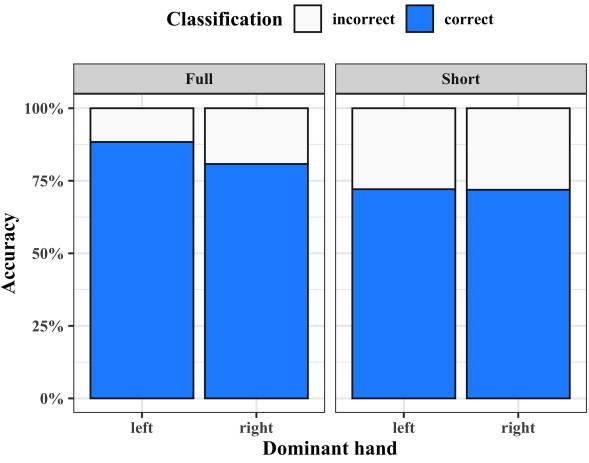

Figure 5: Accuracy of hand dominance estimation with full and short methods.

Figure 6 shows the accuracy of hand dominance estimation across different sign types with regard to the number of hands articulating: one-handed signs (1h) and two-handed signs (2h; unbalanced and balanced). The full method performs better across all three sign types, but unsurprisingly the balanced two-handed signs are approximately at chance level for both methods. The reason for this is that balanced two-handed signs are gener-

ally symmetrical in terms of both hands articulating either mirrored or alternating movements, and the hands would thus be expected to have approximately the same total distance traveled. Consequently, defining hand dominance is less important for balanced signs, since the two hands are generally symmetrical.

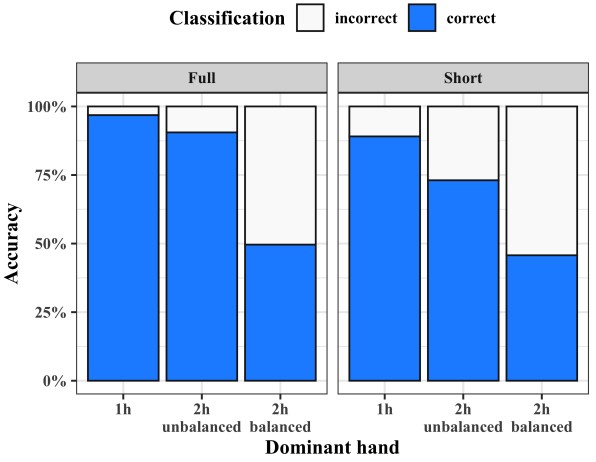

Figure 6: Accuracy of hand dominance estimation with full and short methods by sign type.

### 3.3 Number of Hands

The number of hands involved in each sign video was estimated by comparing the relative distance traveled between the two hands to see whether one hand traveled farther than the other hand by a factor between 1 and 4. In a previous study using OpenPose data, Östling et al. (2018) used a factor of 3 to estimate the number of hands (whether one- or two-handed). Here, the factor is increased by 0.1 increments to evaluate what the best cutoff point is for this data set. Figure 7 shows the $F_1$ scores for one- and two-handed signs across all factor increments for both methods, with the mean $F_1$ as a thicker, black line. The figure demonstrates that the best performing factor is 1.7 for the full method and 1.8 for the short method, and that the full method once again performs better overall. Table 3 shows the accuracy of classification for the best performing factors for each method.

Figure 8 shows a confusion matrix of the classification of one- and two-handed signs across the three sign types: one-handed and two-handed (unbalanced and balanced). Both methods perform relatively well with one-handed signs and balanced two-handed signs, but the unbalanced two-handed signs are particularly problematic for the

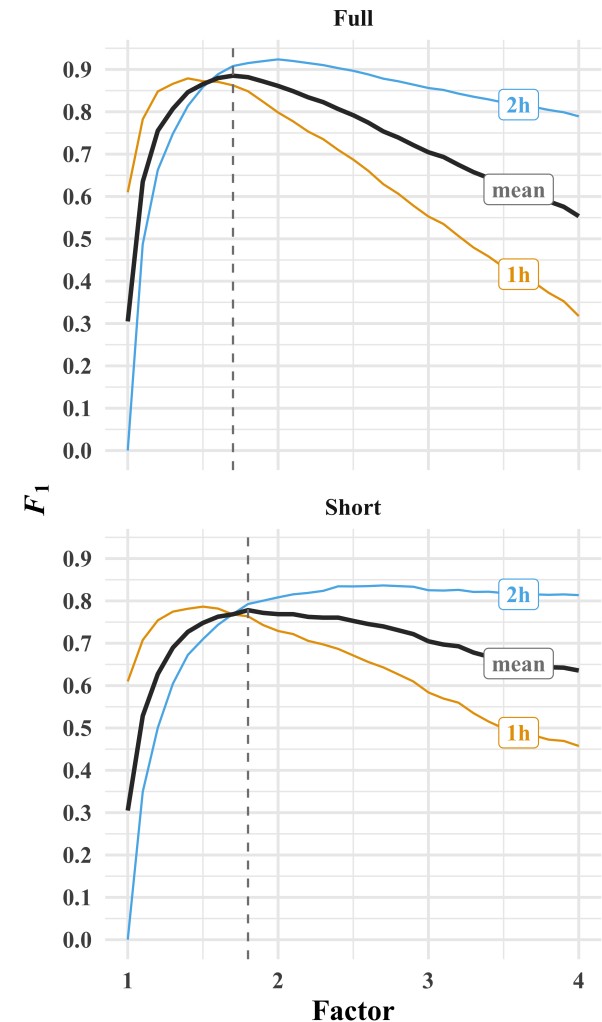

Figure 7: $F_1$ of number of hands estimation with full and short methods. Yellow line shows one-handed signs only, blue line shows two-handed signs only, black line shows the combined mean. Dashed, vertical black line shows the top performing factor for each method.

short method. It is unsurprising that this category poses some problems, seeing as it is an in-between sign type phonologically (cf. van der Hulst, 1996; Sandler, 2006; Crasborn, 2011), in that it has a single hand actively articulating (like one-handed signs) but two hands involved in the sign (like balanced two-handed signs).

### 3.4 Place of Articulation

The place of articulation of the signing for each sign video was estimated using three methods: the full method, including the mean coordinates of the estimated dominant hand across all sign frames; the short method, including the mean co-

| Method | # | Fct | Precision | Recall | $F_1$ |
|--------|-----|-----|-----------|--------|-------|
| Full | 1h | 1.7 | 0.89 | 0.84 | 0.86 |
| Full | 2h | 1.7 | 0.89 | 0.93 | 0.91 |
| Short | 1h | 1.8 | 0.78 | 0.75 | 0.76 |
| Short | 2h | 1.8 | 0.78 | 0.81 | 0.79 |

Table 3: Precision, recall and $F_1$ of number of hands estimation with full and short methods using the top performing factor for each method.

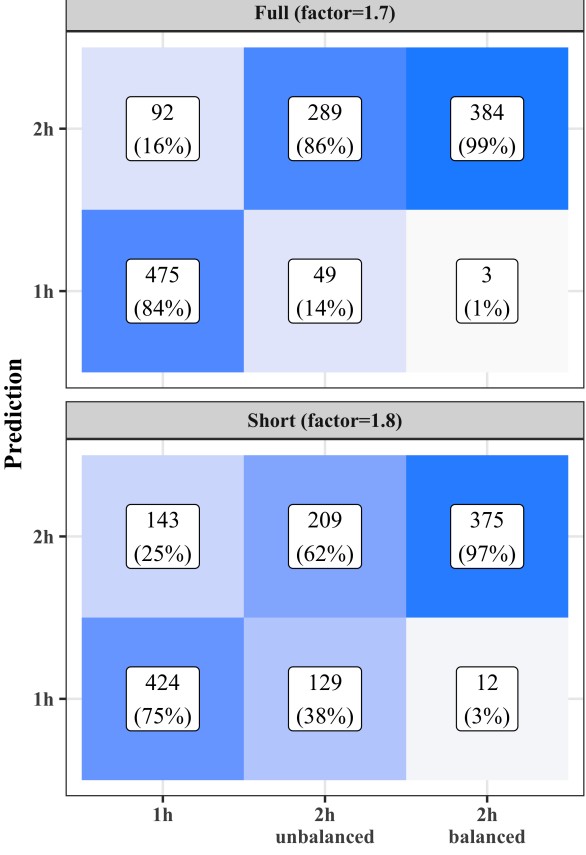

Figure 8: Confusion matrix of number of hands estimation with full and short methods, with absolute numbers and accuracy (%) for each category.

ordinates of the estimated dominant hand only for the frames inside the estimated articulation phase; and the start method, including the coordinates of the estimated dominant hand only for the estimated start frame, i.e., the first inverted peak (sign hold) inside between the transport movement peaks. Figure 9 shows the location of the estimated dominant hand relative to the signer's body across the known places of articulation for the three methods. The figure illustrates that the short and start methods perform much better than the full method. The full method conflates the hand location across the entire sign video, which means that rest positions and transport movements will always be included, and thus the estimated places of articulation are quite uniform across the actual locations as coded in the lexical database. With the short and start methods, there are visible differences in the estimated places of articulation across actual locations, which also reflect the actual locations of the signs in the lexical database – e.g., signs with a known place of articulation by the head are visibly higher up than the others. This pattern is also visible in Figure 10, which simplifies the comparison by looking at the height of the estimated place of articulation. Here, there is a much clearer – and accurate – difference across the known sign locations, showing that the short and start methods outperform the full method.

## 4 Conclusions

In this paper, I have shown initial explorations of methods to extract basic information about articulation and sign form from sign language video data using MediaPipe.

The first step of estimating an approximate articulation phase of the sign proved to be possible for most sign videos in the data set, which turned out to be a fruitful endeavor in order to then accurately estimate the place of articulation across signs. For the purpose of estimating hand positions corresponding to a phonological place of articulation, estimating the articulation phase is crucial, since the signal is otherwise disrupted by noise from rest positions and transport movements. Being able to automatically segment the articulation phase of signs would have other obvious applications, when extracting phonological information about the actual sign (articulation) rather than contextual noise (transport and rest).

However, when estimating hand dominance and number of signs articulating, the full method, which included data from all frames in the sign video, consistently outperformed the short method, for which the data only included frames within the estimated articulation phase. It seems as though the crude method of comparing the relative distance traveled between the two hand benefits from more data than the short articulation phase provides, and that the transport movements to and from the articulation phase are in fact quite useful for magnifying the differences in distance

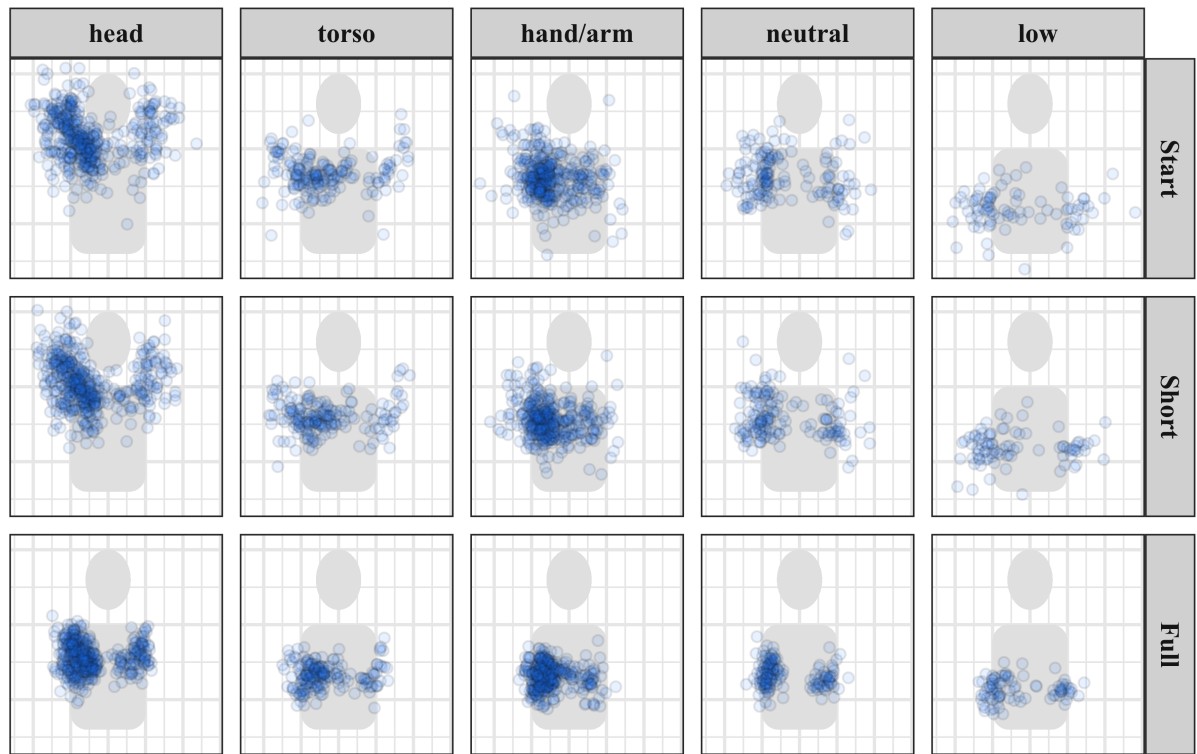

Figure 9: Estimated place of articulation across locations and three methods.

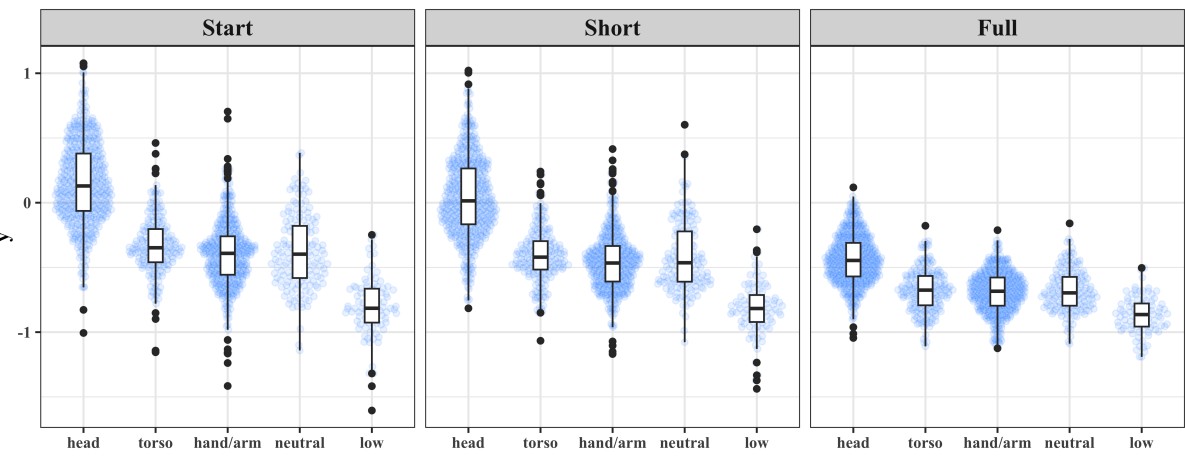

Figure 10: Estimated place of articulation as vertical sign height (*y* coordinates) across locations and three methods.

traveled between the two hands. This method works quite well with dictionary data here, with each video containing a single (non-compound) sign. If applied to complex/compound signs or stretches of multiple signs in succession, as in conversational data, transport movements may not be as distinct and more elaborate methods to estimate articulation phases would be necessary.

The results of this preliminary and exploratory study has demonstrated some possibilities in extracting sign language articulation from videos with MediaPipe, which can be used as a fast and cost-efficient way to analyze pre-recorded but unannotated sign language data in substantially larger quantities than would be feasible with manual annotation.

## Acknowledgments

Thanks to Thomas Björkstrand for sharing Swedish Sign Language dictionary data.

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
