# OpenReview forum: "Extracting Sign Language Articulation from Videos with MediaPipe"
_NoDaLiDa/2023/Conference — NoDaLiDa 2023_

### Official Review · Reviewer_nAUj · 2023-03-06
**-**

**Rating:** 8
**Confidence:** 3

**Review:**

The paper describes the detection of some of the key aspects of sign language articulation. The article is well written and understandable even for readers (like myself) who are not experts in sign language. There is a good explanation of the methods and the experiments show good results.

**Paper Type:**

Long paper

---

### Official Review · Reviewer_TBYo · 2023-03-17
**A well-written article.**

**Rating:** 7
**Confidence:** 4

**Review:**

"Extracting Sign Language Articulation from Videos with MediaPipe" is a well-written article that provides a clear overview of methods for extracting Sign Language (SL) articulation. The author's research was into simple information extraction from SL articulation methods. The author clearly noted the differences between their approach and existing ones. The main limitation of the article is that the author did not discuss more sophisticated extraction methods that might work better for more complex movements.

**Paper Type:**

Long paper

---

### Decision · Program_Chairs · 2023-03-17

Accept